# The Primary Care Perspective on the Norwegian National Strategy against Antimicrobial Resistance

**DOI:** 10.3390/antibiotics9090622

**Published:** 2020-09-19

**Authors:** Guri Rørtveit, Gunnar Skov Simonsen

**Affiliations:** 1Department of Global Public Health and Primary Care, University of Bergen, 07103 Bergen, Norway; 2Department of Microbiology and Infection Control, University Hospital of North Norway, 9019 Tromsø, Noway; gunnar.skov.simonsen@unn.no; 3Faculty of Health Sciences, University of Tromsø—The Arctic University of Norway, 9019 Tromsø, Norway

**Keywords:** antimicrobial resistance, primary care, strategy process

## Abstract

A national strategy to combat antimicrobial resistance (AMR) has been subject to cyclic processes in Norway since 1998. In 2020, a renewed process cycle was launched. Here, we describe the process and the approach of the process. In addition, we describe two concepts from philosophy of science that may help to frame the process: AMR is an example of a super wicked problem, and post-normal science provides tools to analyze the problem from a new angle.

## 1. Introduction

The international struggle against antimicrobial resistance (AMR) is a matter of huge importance, with no simple solutions in sight [1,2]. Over the years, since the usefulness of antibiotics was discovered and large-scale production was developed, antibiotics have been used in healthcare, veterinary medicine, agriculture, fish farming, and other areas. Although AMR is a naturally occurring phenomenon, the wide use of antibiotics is known to drive and spread AMR, which is a problem at the ecologic as well as individual level [1,3]. The need for antibiotics in healthcare cannot be overestimated [4]. They are used to treat deadly infections as well as prevent serious complications after surgery. However, they are also used for less severe infections and even viral infections, where they should not play a role.

The increasing challenge with resistant microbes can reasonably be compared with the climate change problem. Although the scientific basis and the political implications of AMR and climate change are different, there are many commonalities in terms of complexity, the difficulties finding solutions, and the acuteness of the problem. Furthermore, like climate change, AMR is a global problem that cannot be solved by individual countries alone. Still, initiatives and actions at the national level are needed, and engagement from governmental bodies is part of the solution.

Within human medicine the general current approach is to prevent infections, improve appropriateness of antimicrobial use through antibiotic stewardship, and improve sanitation and hygiene. In countries with a strong primary healthcare sector, most infections are managed at that level. Hence, a substantial proportion of antibiotic prescriptions are issued by primary care physicians, implying that this part of the healthcare sector needs to be heavily involved in any strategy to reduce AMR.

In 1998, Norwegian governmental institutions started a national strategy process to combat AMR, with revision and renewal every five year. Currently, such a revision process is ongoing; hence, the content of the new strategy is not yet ready to be published. The aim of the current paper is to describe the strategy process with emphasis on the relevance for the primary healthcare sector. We also analyze the process by use of two concepts from philosophy of science; wicked problems and post-normal science.

## 2. The Norwegian Strategy Process

One of the first steps towards a national strategy was taken in 1998, when the Norwegian Ministry of Health and Social Affairs appointed a project group with the mandate to establish a coordinating plan to fight AMR. The group delivered its report including an action plan in 1999. The action plan resulted in the establishment of the Norwegian Surveillance System for Antibiotics Resistance in Microbes in 2000 and the Norwegian Prescription Database with data available from 2004. Since then, Norwegian health authorities and government have hosted a cyclic strategic process against AMR, directed at the national and international level. The current strategy is based on a report from a multidisciplinary expert group, published in 2014 [5]. The National Strategy against Antimicrobial Resistance (2015–2020) was then released by four ministries of the national government [6]. The four ministries involved (Health and Care Services; Trade, Industry and Fisheries; Agriculture and Food; Climate and Environment) reflected the multidisciplinary competence needed in this effort. The strategy relates to the aims of the World Health Organization’s Global Action Plan on AMR from 2015 [4]. The Norwegian document introduced four overarching goals for the 5-year period (Box 1). The specific goals for the health sector are presented in Box 2. Following up on the strategy, the Ministry of Health presented the National Action Plan in 2015 [7], which launched concrete advice to achieve the goals of the strategy, structured and directed towards national health authorities, the population, primary care (including physicians, dentists and institutions), and secondary healthcare.

Box 1The overarching goals of the current national strategy to reduce AMR (2015–2020) [6].
Reduction of total use of antibioticsMore correct use of antibioticsIncrease knowledge about driving forces and spread of AMR resistanceEngage in international collaboration to strengthen access, correct use and development of new antibiotics, vaccines, and better diagnostic tools


Box 2The specific goals for the healthcare sector in the current national strategy to reduce AMR (2015–2020) [6].
Antibiotics use in the population reduced by 30% (defined daily dose (DDD)/1000 inhabitants/day) compared to the 2012 levelNorway among the three lowest prescribing countries in EuropeAverage prescription of antibiotics reduced from 450 to 250 per 1000 inhabitants per yearPrescriptions of antibiotics for respiratory infections reduced by 20% compared to the 2012 levelConduct studies of AMR burden of disease, consequences of too little use of antibiotics and effects of infection control


In 2020, a new multidisciplinary expert group was appointed by the national government, again with the mandate to lay the basis for a renewed strategy against AMR. The group is specifically asked to update relevant knowledge from 2014, with emphasis on knowledge of strategic relevance. The current group has expertise from the healthcare sector (including primary and secondary levels), microbiology, veterinary medicine, ocean, agriculture, and the wildlife/environment sector. The strategy process has consistently aimed to apply a One Health perspective, which implies a collaborative effort to attain optimal health for people, domestic animals, wildlife, plants and the environment [8]. The group is working to submit our report in the fall of 2020, with an expected follow-up process resulting in a renewed strategy and action plan from the Norwegian government. 

## 3. The Relevance of the Strategy for Norwegian Primary Healthcare

Norway is organized with a strong primary care sector based on general practitioners (GPs) who also run out-of-hours services, a home care system and municipal care institutions for fragile elderly people. Generally, all Norwegian citizens are registered with a GP, who is their primary contact for any medical encounter. The prescription rate for antibiotics is generally low, and AMR is still a relatively minor concern, although increasing [9]. Three of the specific goals for the healthcare sector in the current strategy are directly relevant for primary healthcare (Box 2), including reducing antibiotics use by 30%, reducing the number of prescriptions from 450 to 250 per 1000 inhabitants, and reducing prescriptions for respiratory infections by 20%—all compared to the 2012 level.

By 2020, these goals were completely or nearly reached, and Norwegian primary care physicians should be proud of their contribution. The achievement is even more impressive as the antibiotic prescribing rate always has been low in Norway. Whether this success may be ascribed to the national strategy alone is debatable. There was a trend towards less prescribing before the strategy was launched, and this is part of an international trend in comparable countries such as Nordic countries and the UK [10,11,12]. However, the strategy has undoubtedly supported, and likely also enhanced, this positive ongoing trend.

Two aspects for further reduction of antibiotics prescribing rates should be thoroughly discussed in the upcoming process: firstly, avoiding so-called unnecessary prescribing, and secondly, the balance between low prescribing rate and risk for adverse patient outcomes [13,14,15]. These aspects are particularly relevant for primary care physicians, who generally manage less severe infections, often of viral origin. Current diagnostic tools are not sufficiently precise and safe to decide a priori which patients will develop a severe condition. Hence, prescriptions which turn out to be “unnecessary” are a matter of hindsight not available at the time of the consultation. The acceptable level of uncertainty is a verbal and non-verbal negotiation between doctor, patient, relatives and society at large [16]. A low prescription rate will inherently lead to more cases with adverse outcomes. This must not be a responsibility for the individual doctor to bear but part of a common understanding that some suffering in patients today will result in less suffering for future patients if we can keep antibiotics effective.

An important question for the current expert group is what strategies are purposeful for Norwegian primary care in a situation with an already low prescription rate. There is room for improvement in terms of reducing the number of frequent prescribers, increasing the proportion of narrow spectrum antibiotics, reducing the duration of courses etc. [17,18]. However, applying new targets merely amplifying current targets (Box 2) will probably be neither useful nor achievable. Both primary care physicians and the general population may question the safety and legitimacy of “ever lower antibiotic prescription” and thus jeopardize the strong alliance between healthcare providers, public health authorities and the population, which has been a key success factor in the present strategy.

## 4. Underlying Conditions for a Renewed Strategy

The AMR problem has been described as an example of a “super wicked problem” [19]. “Wicked problems” is a term which was first coined by Rittel and Webber [20] to denote social problems that are so multi-faceted and refractory to solutions that we can only hope to minimize their negative effects. In addition to the original ten criteria of Rittel and Webber, AMR has been suggested to also demonstrate the characteristics of a so-called “super wicked” problem [21]:

(1) The time to solve the problem is running out; (2) Those who are responsible for solving the problem are also partly creating the problem; (3) The mandate for responsible agencies to act is weak; (4) Political acts are not consistent with the need for response. Although, from a legal perspective, the government could assign agencies with a mandate, it is still unclear as to how this could be executed. Societal problems within this category are so complex and have so many different causes that we cannot expect to solve them, but we should nevertheless do our best to minimize their negative effects. The concept of a problem being “super wicked” should not be confused with any problem which is serious or difficult to solve. One may argue that, e.g., the present COVID-19 pandemic is a dramatic example of a very serious global health challenge; still, the emergence and handling of a novel virus and the possible development of a vaccine represent a relatively simple conceptual framework.

The scientific approach to most problems is consequently necessary and useful; however, it is not sufficient to address super wicked problems. Although it does not provide a solution, a useful way of thinking in situations with fundamental uncertainty and an urgent need to act has been developed by Funtowicz and Ravetz [22,23]: the concept of “post-normal science” provides a tool to analyze complex problems in the interface between science and policy, of which AMR is an example. In such situations, facts are uncertain, the stakes are high, values are under dispute, and decisions are urgent. That *facts are uncertain* implies acknowledging that there is a real difference between questions that can be solved by more research and inherent (clinical) uncertainty, where reliable predictions cannot be made now or in the future [24]. For primary care physicians, the latter is the case in many consultations, typically in encounters with elderly patients with respiratory infections. Who will and will not develop deadly complications may be unpredictable in the early phase of any infection. *Stakes are high* means that the outcomes may be successful or devastating depending on the decisions made, and this is difficult to judge beforehand. We do not know how severe the AMR consequences will be in the future, whether we act this way or the other [25,26]. In the primary care context, substantially reducing antibiotics prescriptions may severely harm many people for the potential benefit of future patients, but we do not know the full consequences of our choices, neither good nor harmful. *Values under dispute* may be of political, moral or economic character, implying that two people may view the same situation differently: for a patient, the possibility of relief of symptoms will probably outcompete the consideration for consequences of AMR for the rest of the society. On the other hand, this may be a constant worry for the primary care physician. Countries with different sets of cultural dimensions have been shown to struggle differently with inappropriate antibiotic prescriptions [27]. Finally, and maybe most importantly, *decisions are urgent*, and we cannot wait for science to establish the facts before we act. Professor Peter Gluckman, the former chief science adviser in New Zealand, has launched a set of principles for science advice to the government, based on a post-normal science strategy [28]. These may apply to the work with national and international AMR strategies as well. Neither politicians nor scientists have the solutions, and we must work together and negotiate acceptable steps along the way to reduce the negative impact of AMR. We believe that for an AMR strategy to achieve bold goals, collaboration and respectful interaction between policy makers, political leaders, clinical providers and the scientific community are necessary. The One Health perspective on the AMR problem is internationally accepted, and this implies the need for a broad, multidisciplinary approach both within and among nations. The Norwegian national strategy against AMR is funded on this understanding. The process has been successful this far, in terms of reaching set goals. However, further national reduction of antibiotic prescriptions is not a solution to the global AMR problem and maybe not even to the AMR problem within our own borders.

In conclusion, perceiving AMR as a super wicked problem may help in understanding its scope. Applying post-normal science analytic tools may be useful to address the specific challenges that primary care is faced with internationally within the AMR context. Traditional strategies must be accompanied by new approaches and perspectives. Although the Norwegian strategy so far has been a success in terms of reducing antibiotic prescriptions, this is only a surrogate endpoint. To reach the real goal of reducing AMR, we need a global, multidisciplinary, innovative approach involving all stakeholders. There is no simple solution, and the struggle will be endless, but we still cannot give up.

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
