# Peer review of "The Primary Care Perspective on the Norwegian National Strategy against Antimicrobial Resistance"

_antibiotics, 2020, doi:10.3390/antibiotics9090622_

Round 1

Reviewer 1 Report

The authors have provided a good primary care perspective on the national strategy against AMR in Norway. Thank you for your contribution to this important public health / infectious disease topic.

MINOR COMMENTS:

  • Lines 75-77: Consider expanding on this section to share how the goals were determined? Defined daily dose (DDD) cited in line 231 (but not defined elsewhere) – consider a more thorough explanation for metrics. Any thoughts on using Days of Therapy (DOT) as an alternate metric?
  • Line 99: “specter” misspelling – spectrum?
  • Lines 99-100: grammar issue with the sentence - “…will probably not be useful nor achievable.”
  • Line 106: Consider providing more background or reference re: “wicked problem” as some readers may not be familiar with the use of this terminology
  • Line 147: “between nations” – consider “among nations”?

Author Response

Our response in italic below

Reviewer 1

The authors have provided a good primary care perspective on the national strategy against AMR in Norway. Thank you for your contribution to this important public health / infectious disease topic.

MINOR COMMENTS:

Lines 75-77: Consider expanding on this section to share how the goals were determined? Defined daily dose (DDD) cited in line 231 (but not defined elsewhere) – consider a more thorough explanation for metrics. Any thoughts on using Days of Therapy (DOT) as an alternate metric?

We, the authors, were not part of the previous strategy process and therefore have no access to the discussions underlying the decisions made about goals. DDD is now spelt out. The use of metric in the previous strategy (which is the one cited in the boxes) is not interchangeable. We have clarified in the heading of the two Boxes.   

Line 99: “specter” misspelling – spectrum?

Corrected

Lines 99-100: grammar issue with the sentence - “…will probably not be useful nor achievable.”

Corrected

Line 106: Consider providing more background or reference re: “wicked problem” as some readers may not be familiar with the use of this terminology

We have added text setting the concept of “wicked problems” in a historical frame. We have also included two more references, which give further context to both “wicked” and “super wicked problems”.

Line 147: “between nations” – consider “among nations”?

Corrected

Reviewer 2 Report

The manuscript (commentary) is well written and only requires minor revision. I do have a few content-related and editorial comments.

Lines 25-26, I am not sure whether the challenges of AMR and climate change are that much comparable beyond complexity. Climate change is a constant natural phenomenon over time, the climate is never stable/the same (at least in time scales of hundreds/thousands of years), the dispute is about the impact of mankind on the extent of climate change and the right measures to take which could make the difference (distinction of technically-scientific sound measures and e.g. the current media-mediated CO2 hysteria) . AMR also occurs naturally, but the misuse and even abuse of antibiotics (inappropriate applications) has created this “super wicked problem”.

Line 48, there’ s only one government (or the authors mean the governments in the counties)?

Line 80, ascribed to

Line 86, so-called unnecessary prescriptions

Line 90, Hence, prescriptions which turn out to be unnecessary are a matter

Lines 95 and 131, so the meaning is that being more restrictive with prescriptions of antibiotics today will reduce AMR in the future and thus make suitable use of antibiotics more effective?

Line 97, with an already

Line 100, will probably be neither

Line 108, from a legal perspective, the government could assign agencies with a mandate, the question is more how this could be executed

Line 124, reliable predictions?

References, please adhere to the formatting rules of the journal: Author 1, A.B.; Author 2, C.D. Title of the article. Abbreviated Journal Name Year, Volume, page range.

Author Response

Our response to reviewer 2 in italic below.

Reviewer 2:

The manuscript (commentary) is well written and only requires minor revision. I do have a few content-related and editorial comments.

Lines 25-26, I am not sure whether the challenges of AMR and climate change are that much comparable beyond complexity. Climate change is a constant natural phenomenon over time, the climate is never stable/the same (at least in time scales of hundreds/thousands of years), the dispute is about the impact of mankind on the extent of climate change and the right measures to take which could make the difference (distinction of technically-scientific sound measures and e.g. the current media-mediated CO2 hysteria) . AMR also occurs naturally, but the misuse and even abuse of antibiotics (inappropriate applications) has created this “super wicked problem”.

Interesting comment, although we uphold our view that climate change is comparable to AMR seen in the perspective of super wicked problems. To clarify, we have revised the text in line 26-27. 

Line 48, there’ s only one government (or the authors mean the governments in the counties)?

There have been several national governments in Norway since the strategy process started (1998), with changing parties involved. However, in terms of government as an institution, there is only one. We have changed this to singular.

Line 80, ascribed to

Corrected

Line 86, so-called unnecessary prescriptions

Corrected

Line 90, Hence, prescriptions which turn out to be unnecessary are a matter

Corrected

Lines 95 and 131, so the meaning is that being more restrictive with prescriptions of antibiotics today will reduce AMR in the future and thus make suitable use of antibiotics more effective?

This is now clarified in line 97-98 (line 95 in the former manuscript)

Line 97, with an already

Corrected

Line 100, will probably be neither

Corrected

Line 108, from a legal perspective, the government could assign agencies with a mandate, the question is more how this could be executed

This point has been included in line 117-118

Line 124, reliable predictions?

Corrected

References, please adhere to the formatting rules of the journal: Author 1, A.B.; Author 2, C.D. Title of the article. Abbreviated Journal Name Year, Volume, page range.

Corrected